

# Non-destructive sub-surface inspection of multi-layer wind turbine blade coatings by mid-infrared Optical Coherence Tomography

Coraline Lapre[1], Christian Rosenberg Petersen[1,2], Per Nielsen[3], Thomas Wulf[3], Jakob Ilsted Bech[4], Søren Fæster[4], Ole Bang[1,2,5], and Niels Møller Israelsen[1,2]

[1]DTU Electro, Department of Electrical and Photonics Engineering, Technical University of Denmark, Ørsteds Plads 343, Kgs. Lyngby, 2800, Denmark
[2]NORBLIS ApS, Virumgade 35D, Virum, 2830, Denmark
[3]Siemens Gamesa Renewable Energy A/S, Assensvej 11, Aalborg, 9220, Denmark
[4]DTU Wind, Department of Wind and Energy Systems, Technical University of DenmarkFrederiksborgvej 399, Roskilde, 4000, Denmark
[5]NKT Photonics A/S, Blokken 84, Birkerød, 3460, Denmark

**Correspondence:** Coraline Lapre (corla@dtu.dk)

**Abstract.** Non-destructive inspection (NDI) is useful in the industrial sector to ensure that manufacturing follows defined specifications, reducing the quantity of waste and thereby the cost of production. Optical Coherence Tomography (OCT), a well-known diagnostic technique in medical and biological research, is increasingly being used for industrial NDI. In the mid-infrared (MIR) wavelength range, OCT can be used to characterise parts and defects not possible by other industry-ready

scanners, and enables better penetration than conventional near-infrared OCT.

In this article, we demonstrate NDI of wind turbine blade (WTB) coatings using an MIR OCT scanner employing light around $4\,\mu$m from a supercontinuum laser source. We inspected the top two layers of the coating (topcoat and primer) in two different samples. The first is to determine the maximum penetration depth, and the second one is to imitate defect identification. The results of our study confirm that MIR OCT scanners are promising for coating inspection and quality control in the

production of WTBs, with performance parameters not achievable by other technologies.

## 1 Introduction

In recent years, the development of renewable energy sources has greatly intensified, in part due to the ongoing transition away from fossil fuels. For countries like Denmark, where coastal winds are powerful and constant, developing new offshore wind turbines is essential. However, the typical lifespan of wind turbine technology is limited to around 20 years, or up to 25 years

in optimal conditions, which is mainly due to erosion on the blades (Lichtenegger et al. (2020); Beauson et al. (2022)).

One of the critical challenges in manufacturing wind turbine blades (WTB) is achieving durable and high-quality coatings. These coatings are vital for protecting the blade's internal structure and mitigating delamination effects, especially on the leading edge (Slot et al. (2015); Mishnaevsky et al. (2021); Fæster et al. (2021); Mendonça et al. (2023)). In industrial contexts,

Non-Destructive Inspection (NDI) techniques play a crucial role in ensuring that materials, samples, or parts follow manu-



facturing specifications. The common method used for testing the coating quality in production is a so-called "dolly test", a destructive method where the coating layer is removed in a region of interest and then characterised. This approach is inherently destructive, limited in scope, and cannot provide a comprehensive evaluation of the entire blade.

Different non-contact methods focus on assessing the internal integrity of the WTB. Thermography achieves an average penetration depth of a few centimetres and works well with composite materials such as glass fibre reinforced polymer (GFRP), but only provides a surface projection with millimetre-level resolution (Doroshtnasir et al. (2015); Ardebili and Alaei (2022)). Vibration analysis or acoustic emission techniques enable manufacturers to detect damage or crack propagation at an early stage. They can detect cracks as small as approximately 25 mm, as recent studies have shown. The results are also presented
in the form of surface projections (Yang et al. (2018); Asokkumar et al. (2021)). For deeper inspections, in the order of metre, radiographic methods (e.g. X-ray CT) (Mishnaevsky et al. (2021); Fæster et al. (2021)), and microwave (e.g. super high frequency or terahertz frequency) (Li et al. (2016); Shin Yee et al. (2024)) inspection are effective for detecting internal faults in the blades structure. Radiographic methods provide volumetric scanning, unlike microwave methods, which provide results via surface projection. However, for radiographic methods, as resolution is inversely proportional to the size of the area inspected,
for an entire WTB, the transverse and lateral resolution is often too low to detect a defect inside the coating. Furthermore, the radiation emitted by this method is hazardous to users and could damage the material being characterised.

Another imaging technique well-known in the biomedical field for several decades but less common in the industrial sector is Optical Coherence Tomography (OCT) (Fujimoto et al. (2000); Drexler et al. (2001)), based on combined laser scanning
and subsurface light echoes deduced from interference signals. This technology has the advantage of being fast and suitable for characterising organic materials and achieving ultra-high depth resolution around 1-50 $\mu$m and a penetration depth around 1-2 mm.

The mid-infrared (MIR) OCT scanner operating at approximately 4 $\mu$m, has already demonstrated its capability for non-contact, sub-surface NDI of marine coatings (Petersen et al. (2021)), WTB coatings (Petersen et al. (2023); Lapre et al.
(2024b)), paper quality inspection (Hansen et al. (2022)), credit card inspection (Israelsen et al. (2019)), and ceramic inspection (Zorin et al. (2022); Lapre et al. (2024a)). Because of the use of MIR wavelengths light scattering is reduced, which enables the MIR OCT technology to penetrate several hundreds of microns into even highly scattering coatings with high depth resolution, which includes both the topcoat and primer layers of modern WTBs. The depth resolution can be as low as around
5 $\mu$m because of the use of a unique ultra-broadband supercontinuum laser covering an enormous bandwidth of 1-4.6 $\mu$m. In this article we demonstrate non-contact, sub-surface NDI of the combined topcoat and primer layers of WTB coatings to a depth of approximately 360 $\mu$m using a MIR OCT scanner. We demonstrate that MIR OCT compares favourably with X-ray tomography and we demonstrate an automated algorithm for monitoring the thickness of multiple coating layers, which is based on the scattering and absorption properties of the material.



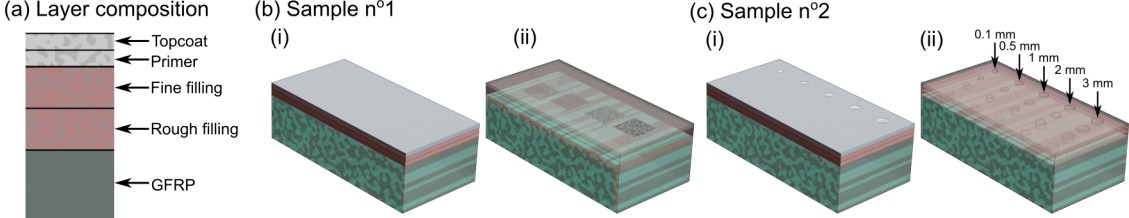

**Figure 1.** (a) Schematic of the layer composition of the samples visualised in (b,c). (b) Sample with aluminium foil patches placed at the interfaces between layers. (c) Sample with punch defects in different layers. For (b-c), (i) and (ii) show opaque and transparent 3D illustrations of the samples.

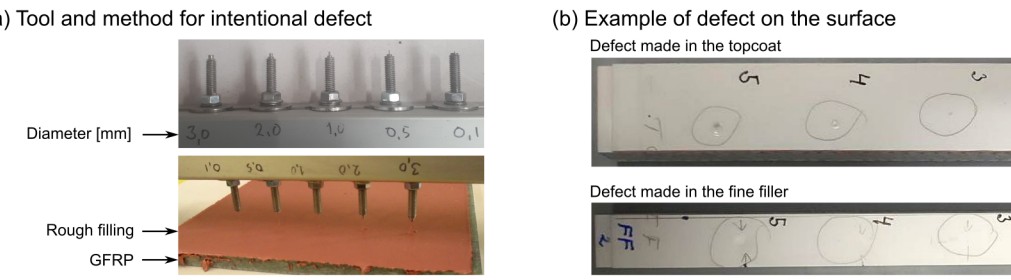

**Figure 2.** (a) The tool and method used to create the defects in each layer during the sample manufacturing. (b) Two example pictures of the surface of the sample with intentional defects in the topcoat and in the fine filler.

## 2 Materials and methods

### 2.1 The blade samples

Two sets of samples were tested, denoted n°1 and n°2, one set with aluminium foil inside, and the other one containing intentional defects. Figure 1(a) shows a schematic of the layer composition above the glass fibre reenforced polymer (GFRP) the material used for the inside of the WTB, and Fig. 1(b-c) presents 3D illustrations of both samples. During the process of fabrication, the layers are subsequently applied and dried in a controlled environment. Each layer is sanded before the application of the next one. For sample n°1, between each layer a small square of aluminium foil was inserted. It was used to benchmark the penetration depth of the MIR OCT scanner and the refractive index of the layers.

For sample n°2, at each layer, before the drying process, some defects were created to imitate some manufacturing defects (cf. Fig 1(c) and Fig. 2).

For each layer, 5 holes with different diameters were made by punching the paint in staggered rows with metallic needles with different diameters: 0.1 mm, 0.5 mm, 1 mm, 2 mm, and 3 mm. The punched holes subsequently undergo two processes before the layer is entirely dry:





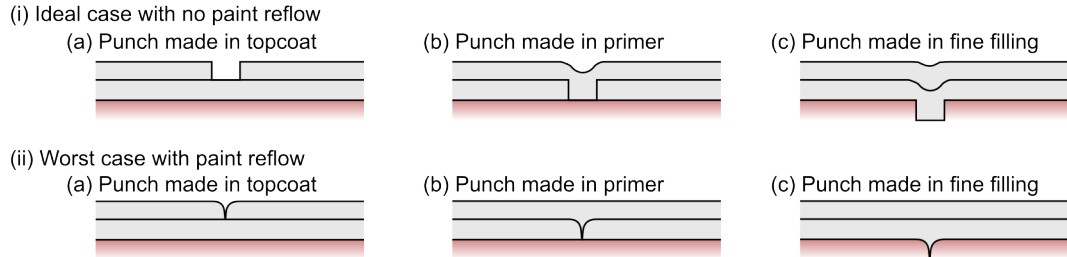

**Figure 3.** Illustration of the impact of introducing punch defects at different layers in the production with (i) an ideal case without any reflow, and (ii) the worst case of reflow of wet paint, (a) in the topcoat, (b) in the primer, and (c) in the fine filling.

(i) In an ideal case, the hole made after the punch doesn't change and keeps the size of the needle. Then the hole is filled with a subsequent paint layer, if it is not made in the top layer.

(ii) In most cases, there is a more or less important reflow of wet paint after punching. This reflow could entirely fill the hole.

The resultant defects after processes (i) and (ii) are illustrated in Fig. 3(i) and Fig. 3(ii) for the three punched layers corresponding to (a), (b), and (c), respectively. The deeper the layer the punch is made, the less severe the disturbance in the surface of the coating is expected. In contrast, punches in the topcoat are not remedied by subsequent coating layer deposition and are most likely to persist. For this reason, punches in the topcoat will only be remedied by process (ii).

The materials used for each layer are as follows:

1. Topcoat CWind UHS Topcoat + Relest Hardener PUR 307 (CLEAR) + Relest Thinner PUR 307.
2. Primer : Carboline Windmastic®400 FC Primer (PART A and B) + Carboline Thinner no.25.
3. Fine filling (Awl 8020) : Awlfaire Pumpable Base + Awlfaire Pumpable Converter.
4. Rough filling (Awl 8200) : Awlfaire LW White Base + Awlfaire LW Fast Converter.

The layer thicknesses of both samples are summarised in Table 1. Siemens Gamesa provided the thickness values for the layers in the wet form, before drying and sanding. The value when the layers are dry were calculated from the datasheet of the paint, the MIR OCT measurements, and the X-ray CT measurements. In a previous study in Petersen et al. (2021), it was demonstrated that MIR OCT has the capacity to measure the thickness of wet paint film and follow the shrinking of the paint during curing; this corroborates the thickness measured from the OCT and X-ray CT.

## 2.2 The OCT scanner

The MIR OCT scanner is presented in Fig. 4(a). The light source is an in-house fabricated MIR supercontinuum (SC) source covering a spectrum from 1 to 4.6 $\mu$m (Woyessa et al. (2021)) coupled to a Michelson interferometer. Due to the range of the spectrometer and to minimise the amount of unnecessary power, there is a long-pass filter at the input of the interferometer, which removes light at wavelengths shorter than 3.2 $\mu$m (cf. Fig. 4). At the output of one arm of the interferometer, the scan-



|  | Layers of sample n°1 | | | |
| --- | --- | --- | --- | --- |
|  | Topcoat [$\mu$m] | Primer [$\mu$m] | Fine filling [$\mu$m] | Rough filling [$\mu$m] |
| Wet film, data provided by Siemens Gamesa | 210 | 290 | 500 | 1050 |
| Dry film calculated with datasheet | $\sim 149$ | $\sim 255$ | 500 | 1050 |
| Dry film measured with OCT above foil | $\sim 144 \pm 2.9$ | $\sim 219 \pm 2.3$ | unknown | unknown |

|  | Layers of sample n°2 | | | |
| --- | --- | --- | --- | --- |
|  | Topcoat [$\mu$m] | Primer [$\mu$m] | Fine filling [$\mu$m] | Rough filling [$\mu$m] |
| Wet film, data provided by Siemens Gamesa | 170 | 190 | 500 | 2000 |
| Dry film calculated with datasheet | $\sim 120$ | $\sim 166$ | 500 | 2000 |
| Dry film measured with OCT | $\sim 120 \pm 2.9$ | $\sim 123 \pm 2.3$ | unknown | unknown |
| Dry film measured with X-ray | $\sim 240 \pm 7.5$ | | $\sim 338 \pm 7.5$ | $\sim 1539 \pm 7.5$ |

**Table 1.** Wet and dry film layers thicknesses in micrometres for the "foil sample" (sample n°1), and for the "defect sample" (sample n°2).

ning is achieved using a two-axis silver-coated mirror galvanometric scanner coupled to an achromatic germanium lens with a focal length of 30 mm. The average power on the sample arm was about 20 mW.

The lens inside the scan head introduces chromatic dispersion, which occurs due to a variation in the speed of light depending on the wavelength as it travels through a material. In response, a germanium window was inserted in the reference arm to compensate the dispersion, the residual dispersion mismatch was compensated numerically (Wojtkowski et al. (2004)). The system achieved an axial (depth) resolution of $\sim 9.74$ $\mu$m in the air, $\sim 6.54$ $\mu$m in topcoat, and $\sim 5.29$ $\mu$m in primer, and a transverse spatial resolution of $\sim 22$ $\mu$m, determined by using a standard USAF-1951 target. The maximum sensitivity was
60 dB and the 6 dB sensitivity roll-off depth was 1.9 mm in air for a 0.3 ms A-scan integration time. For more information about the MIR OCT system, see in Israelsen et al. (2019, 2021). To compare with more commonly available OCT systems, the measurement was also made with a near-infrared (NIR) OCT system with a centre wavelength around 1.3 $\mu$m (cf. Fig. 6(c-d)), which is described in Israelsen et al. (2017, 2018).

### 2.3  Scanning parameters

The scanning parameters for the characterisation of the sample n°1 were the following, 500 cross-sectional images (B-scans) composed of 500 depth scans (A-scans). For sample n°2, 1000 B-scans composed of 1000 A-scans were used. Both sets of scan parameters covered an area of $\sim 5.73 \times 5.73$ mm$^2$, with each A-scan and B-scan being separated by $\sim 11.49$ $\mu$m for the sample n°1, and by $\sim 5.73$ $\mu$m for the sample n°2. Each A-scan took 2500 $\mu$s to be acquired. As the lens adds curvature to





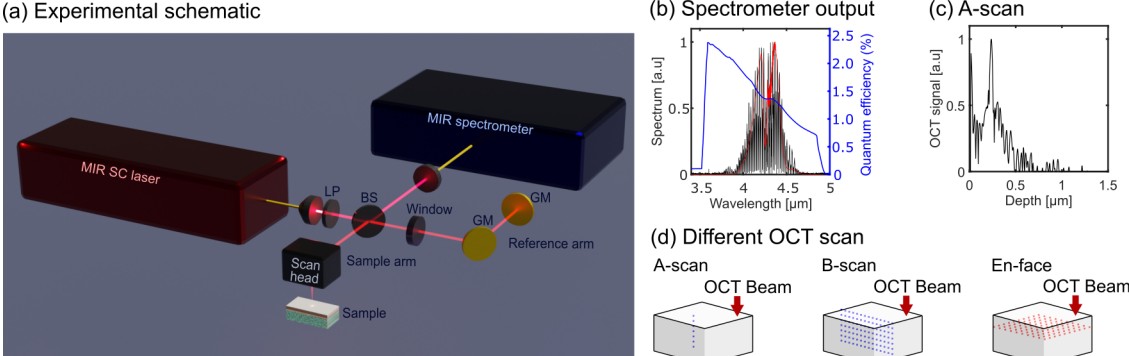

**Figure 4.** (a) 3D representation of the MIR OCT system, SC: supercontinuum, LP: 3.2 $\mu$m long pass filter, BS: beam splitter, GM: gold mirror. (b) Reference spectrum in red, measured interference spectrum in black, upconversion efficiency in blue, and (c) the corresponding calculated A-scan. (d) Schematic of the different OCT scans used in this article.

the cross-sectional image, a flattening post-processing procedure was applied for correcting this effect. The different types of OCT scans presented in this article are illustrated in Fig. 4(d).

### 2.4 Thickness estimation

A beam propagating inside a turbid medium is attenuated because of scattering and absorption. In the case of light propagating in a homogeneous medium, the irradiance $L(z)$ [W cm$^{-2}$] of the beam follows Beer-Lambert's law:

$$L(z) = L_0 e^{-\mu z} \tag{1}$$

with $L_0$ the incidence irradiance and $\mu$ the attenuation coefficient Vermeer et al. (2013); Li et al. (2020). Extraction of the attenuation coefficient from OCT measurements is done by fitting the exponential function from equation 1 to the OCT signal as a function of depth, the A-scan. As OCT images are most depicted on a logarithmic signal form, an exponential decay appears as a linear decay in the OCT A-scans. The presence of noise and speckles inside our data requires averaging over more than a dozen A-scans to enhance the precision in determining the different linear signal decay rates in the different layers Bashkansky and Reintjes (2000); Desjardins et al. (2007).

We developed a primitive automatic interface calculation program based on fitting the attenuation coefficient of the different coating layers. To develop this algorithm, we first calculated the *objective fits* for the decay of the different layers outside any intentional defect region. As the coating layers are not totally homogenous and to reduce noise, speckle, and potential non-intentional defect fluctuation, we calculated a smooth B-scan from an the average of 100 neighbouring B-scans. Then, we calculated an averaged A-scan profile on the entire size of the smooth B-scan, and the decay rate *objective fits* were determined.





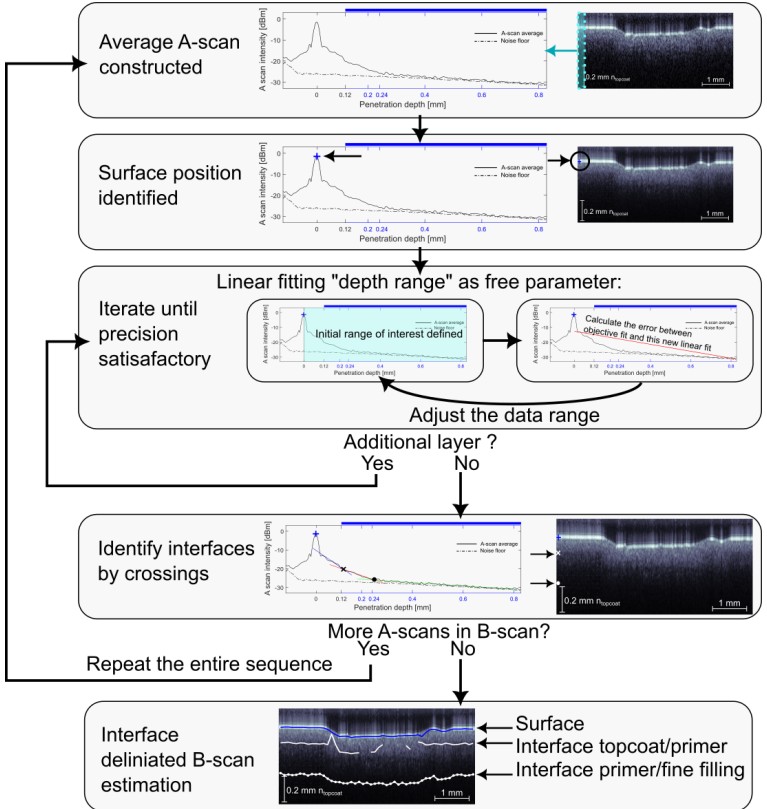

**Figure 5.** Graphical representation of our algorithm.

After the calculation of this *objective fit*, from an area (hopefully) without defects, we can then run the automatic interface recognition program in the interested area. It is possible to run the program with no averaging B-scan, but to enhance the automatic recognition of each interface, we averaged 10 neighbouring B-scans around the interested B-scan and then applied the algorithm on it. The algorithm follows the protocol outlined in the Fig. 5

## 3 Results

Evaluation of the capability of MIR OCT for NDI of WTBs follows several steps. First, a comparative NIR OCT and MIR OCT scanner benchmarking of the penetration depth is presented. Secondly, imitated WTB defects are evaluated for MIR OCT against X-ray CT images. In continuation, coating layer delineation is presented, and afterwards an example of sub-surface additional defect categorisation is finally given.





## 3.1 Benchmarking against NIR OCT

Sample n°1, containing interface aluminium foil markers, was used to assess the penetration depth of MIR OCT against conventional NIR OCT and calculated the refractive of topcoat and primer.

The MIR OCT scanner could penetrate both the topcoat and the primer, detecting a clear reflection from the surface and the top two aluminium foils between the topcoat and primer and between the primer and the fine filling. This is observed in Fig. 6(a,b) where respective layer interface foil signals are mapped out. The foil between fine filling and rough filling was not detected with the MIR OCT system, presenting a penetration depth limit in terms of layer interfaces.

The NIR OCT scanner did not have the same depth performance as the MIR OCT scanner. In a quick measurement with the NIR OCT it was not possible to detect the aluminium foil in the B-scan. For this reason, in a small area, the coating was scratched until the foil were visible by eye at the topcoat/primer and primer/fine filling interface. In Fig. 6(c,d), the green dashed line highlighted the region where the coating where removed. The ghost images above and under the aluminium are present because the foil acting like a mirror.

These measurements were used to determine the refractive index of the topcoat and the primer. As $n = OPD/L$, with $OPD$ the Optical Path Delay, $L$ the physical distance, and $n$ the refractive index of the media, it is possible to calculate the refractive index of the topcoat and the primer. The physical distance of topcoat and primer were measured from the NIR OCT B-scan, in the area the coating were removed.

We obtained from the NIR OCT data $L_{\text{topcoat}} = 144\mu m$ and $L_{\text{topcoat+primer}} = 363\mu m$. Then the $OPD$ of the topcoat and the primer were extracted from the MIR OCT data $OPD_{\text{topcoat}} = 215\mu m$, and $OPD_{\text{topcoat+primer}} = 618\mu m$. Through these results, the refractive index of the topcoat and primer are :

$$n_{\text{topcoat}} = \frac{OPD_{\text{topcoat}}}{L_{\text{topcoat}}} \simeq 1.49 \qquad \text{and} \qquad n_{\text{primer}} = \frac{(OPD_{\text{topcoat+primer}} - OPD_{\text{topcoat}})}{(L_{\text{topcoat+primer}} - L_{\text{topcoat}})} \simeq 1.84 \qquad (2)$$

In the following figures including Fig. 6(a-d), for a better understanding, we make the compromise to calculate the depth scale presented in the B-scans with the refractive index from the topcoat $n_{\text{topcoat}} \simeq 1.49$.

Figures 6(e) and (f) present the A-scan average of each B-scan depicted in Fig. 6(a-d) with the $OPD$ scale in (e), and with the scale calculated with $n_{\text{topcoat}} \simeq 1.49$ (from 0 to 0.14 $\mu$m ) and then calculated with $n_{\text{primer}} \simeq 1.84$ (blue part from from 0.14 $\mu$m) in (f). The black curves present the A-scan averages of the MIR OCT B-scan, solid line (a) and dashed line (b), while the two red curves present the A-scan averages of the NIR OCT B-scan, solid line (c) and dashed line (d). The average was made over 100 A-scans, which represent 1.146mm for MIR OCT and 0.29mm for NIR OCT. MIR OCT allows us to characterise the sample through the primer and slightly into the fine filling until around $360\mu m$. For the NIR OCT evaluation, the signal is completely attenuated after a physical depth of $100\mu m$ due to strong scattering. In the NIR, the signal is dominated by multiple scattering, so the real OCT signal penetration depth is most likely significantly shorter Israelsen et al. (2019).





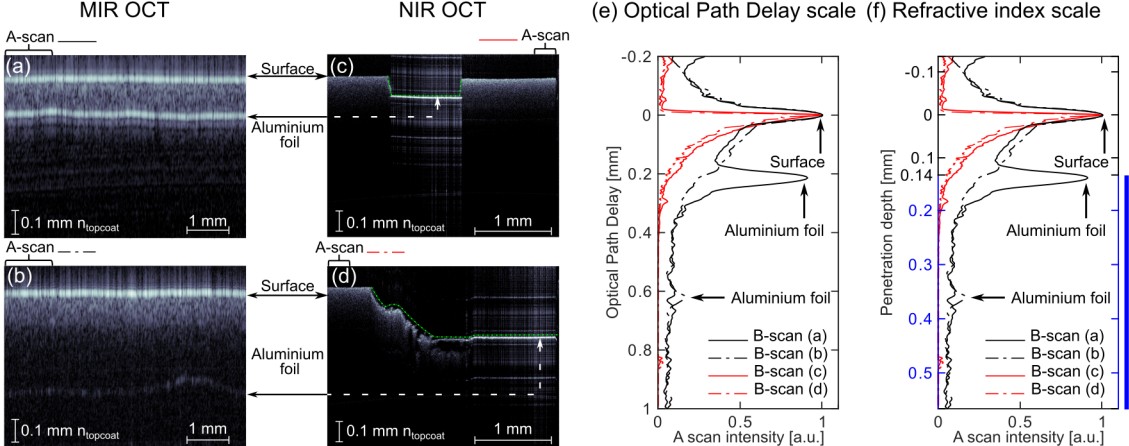

**Figure 6.** (a-b) Comparison of penetration depth between B-scans in MIR OCT (4 $\mu$m centre wavelength) and (c-d) in NIR OCT (1.3 $\mu$m centre wavelength). The dashed green line in (c-d) shows the part where the coating was intentionally removed for the NIR OCT measurement to find the presence of aluminium foil. For each B-scan, the physical distance in depth was calculated with $n_{\text{topcoat}} \simeq 1.49$. (e-f) Comparison of the A-scan average of each B-scan is depicted in (a-d). (e) Present the results with the Optical Path Delay ($OPD$) scale (false scale) and (f) scale with $n_{\text{topcoat}} \simeq 1.49$ (from 0 to 0.14 $\mu$m ) and then scale with $n_{\text{primer}} \simeq 1.84$ (blue scale from 0.14 $\mu$m) (real scale).

|  | Diameter needle | | | | |
|---|---|---|---|---|---|
|  | 3 mm | 2 mm | 1 mm | 0.5 mm | 0.1 mm |
| Topcoat | ✓ | ✓ | ✓ | ✓ | ✓ |
| Primer |  | ✓ | ✓ |  |  |
| Fine filling | ✓ | ✓ |  | ✓ | ✓ |
| Rough filling |  |  |  |  |  |

**Table 2.** Defects characterised by MIR OCT.

## 3.2 Identification of defects by MIR OCT

Sample n$^{\text{o}}$2, made with intentional defects (process explain in section 2.1), was used to imitate the effect of manufactured in the coating quality. The defects identified and characterised by MIR OCT are summarised in Table 2. The highest rate of identified defects was found in the topcoat until the minimum size of the needle. The relation between the identified defects and the needle size is, however, less intuitive in the subsequent layer. More defects were identified in fine filling instead of primer, these observations reflect the way how the punches were made wasn't consistent and manufacturing uncertainty. With

these observations, we acknowledge that punches as small as 100 $\mu$m made in the filler are inflicting changes in the coating.





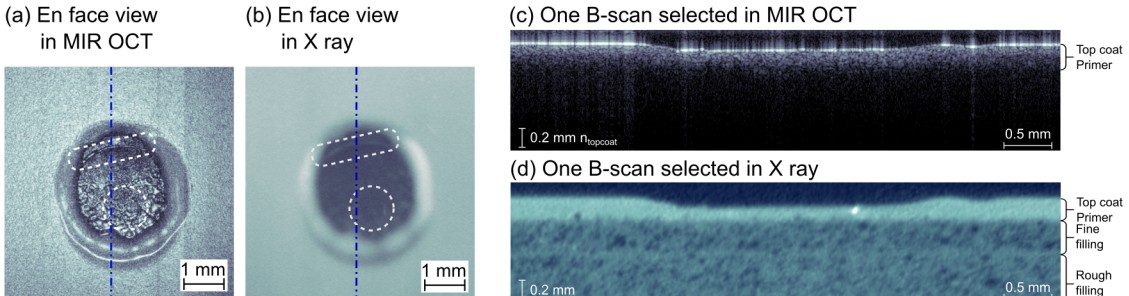

**Figure 7.** Comparison between MIR OCT and X-ray CT measurements made of the defect made in the topcoat part with a diameter of 3 mm. (a-b) *En face* view of the surface of the sample. (c-d) Selected B-scan in approximately the same place. The blue dashed lines in (a-b) show where the B-scans were extracted.

### 3.2.1 MIR OCT compared with X-ray CT

Figures 7-8 show comparisons between X-ray CT and OCT measurements from the sample n°2 made inside the topcoat with a 3mm diameter needle. The X-ray CT machine was a *Zeiss Xradia 520 Versa*, with a resolution of 1-50 $\mu$m in samples of size 1-50 mm. The *En face* view from OCT and X-ray CT is presented in Fig. 7(a,b), respectively, from which it was possible to identify the same areas on the sample, highlighted by the white dashed boxes. The resolution of X-ray CT is lower than OCT due to the large scanning area, chosen to avoid too long scanning time. Indeed, the total time for the characterisation of an area that contains 3 defects, $\sim$10×1×1.5 cm, was around 6hours. For the OCT, the scanning time was 40 min for each defects area $\sim$5.73×5.73 mm$^2$. With the resolution of MIR OCT, it is possible to detect the surface bump.

To further compare the two technologies, we selected one B-scan from MIR OCT and one B-scan from X-ray CT (cf. Fig. 7(c,d)) approximately at the same location, shown by the dashed blue line in Fig. 7(a,b). On both, on the right, the brackets show the different layers position. The penetration depth is better in X-ray CT, however, the issue here is the resolution. This resolution is correlated to the sample size and so scanning area, and the larger the sample, the lower the resolution of the X-ray CT with a fixed time Mishnaevsky et al. (2021). The resolution of OCT in depth depends only on the laser source, and the transverse resolution depends on the scanning head, and for our case, the imaging lens used. The smallest granular visuals in Fig. 7(a) and (c), come from the presence of speckles within the measurements.

To achieve a better comparison between MIR OCT and X-ray CT, we selected the first millimetre horizontally of Fig. 7(c,d) and show this in Fig. 8(b,c). An average A-scan is associated with the B-scan from MIR OCT (cf. Fig. 8(a)), and a pixel average is associated with the B-scan from X-ray CT (cf. Fig. 8(d)). For Fig. 8(a), the spatial average was made over $\simeq$0.991mm (173 A-scan), and for Fig. 8(d), over $\simeq$0.998mm (66 A-scan). In both cases, it is possible to distinguish the topcoat and the primer part, represented by the first two slopes in (a) and the first two plateaus in (d). The third slope in the MIR OCT A-scan plot is the beginning of the fine filling layer, continuing until the noise floor ((black dashed curve) in Fig. 8) at around 0.36mm.





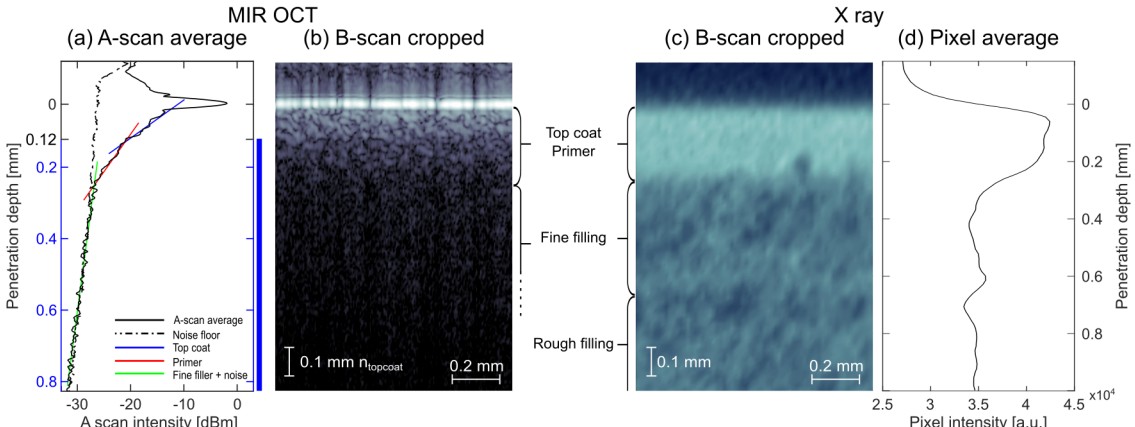

**Figure 8.** Comparison between (b) MIR OCT B-scan and (c) X-ray CT B-scan. (a) is the A-scan average of (b) and (d) is the pixel average of (c). For (a) the penetration depth is calculated with the $n_{\text{topcoat}}$ (from 0 to 0.12 $\mu$m ) and then $n_{\text{primer}}$ (blue scale from 0.12 $\mu$m )). The solid lines blue, red, and green are the fit of topcoat, primer, and fine-filling combine with the noise floor. blue fit $= (-75.09 \pm -9.81) \times OPD/n_{\text{topcoat}} + (6.10 \pm 2.9)$; red fit $= (-47.53 \pm -6.53) \times OPD/n_{\text{primer}} + (-7.25 \pm -0.89)$; green fit $= (-4.31 \pm -0.10) \times OPD + (-23.12 \pm -0.15)$

| | MIR OCT | X-ray CT (Zeiss Xradia 520 Versa) |
|---|---|---|
| Resolution | Not depending on the sample size. Axial : $\sim 9.74$ $\mu$m in the air, $\sim 6.54$ $\mu$m in topcoat, and $\sim 5.29$ $\mu$m in primer. Transverse : $\sim 22$ $\mu$m | Depending on the sample size. Axial and transverse : 1-50 $\mu$m in samples of size 1-50 mm |
| Penetration depth | 360 $\mu$m | Penetrate the entire sample (1.5 cm) |
| Aquisition time | 40 min for an area of $\sim 5.73 \times 5.73$ mm$^2$ | 6h minimum for $\sim 10 \times 1 \times 1.5$ cm |
| Measurement method | Stay normal at the surface of sample | Need to rotate around the sample |

**Table 3.** Comparison summary between MIR OCT and X-ray CT method

Beyond this point, the losses are too pronounced to detect the bottom part of the fine filling. Blue, red, and green solid lines in Fig. 8(a) are, respectively, the fit of the topcoat, primer, and combination of fine filler plus noise.

### 3.2.2 Interface extraction by MIR OCT attenuation coefficient

We conducted a study of coating thickness, topcoat, and primer on sample n°2 (cf. Fig. 9) by tracking the bottom interface of the topcoat and the primer layer. Figure 9(a) shows on the B-scan where are the different average A-scan regions, while 205  Fig 9(b-d) present the corresponding A-scan averages (black solid curves). The average parameters are for (b) 173 neighbouring A-scan ($\sim 0.9913$ mm), (c) 10 neighbouring A-scan ($\sim 0.0573$ mm), and (d) 300 neighbouring A-scan ($\sim 1.719$ mm). The





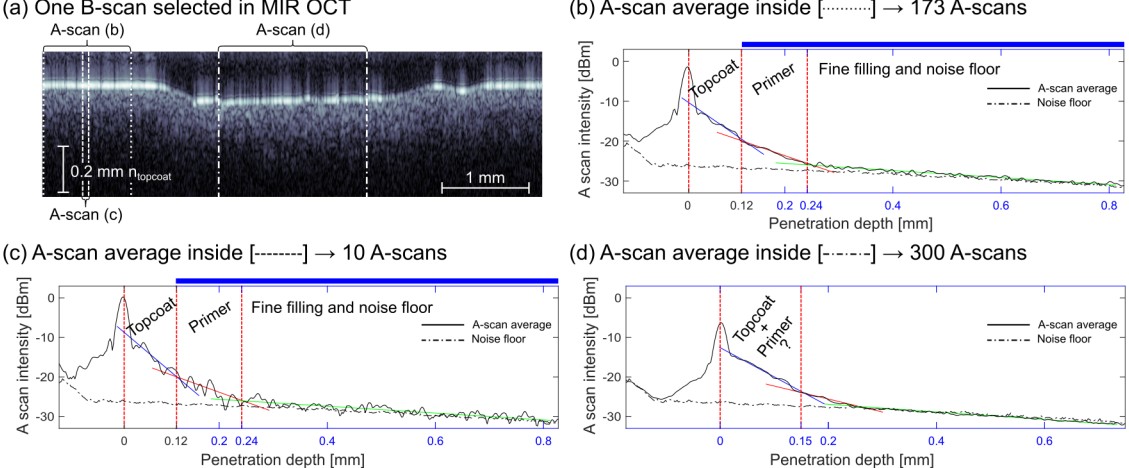

**Figure 9.** (a) The B-scan selected is the same as in Fig. 7, the dashed boxes show where the A-scan was extracted to calculate the average in (b-d). For (b-c) the penetration depth is calculated with the $n_{\text{topcoat}}$ (from 0 to 0.12 $\mu$m ) and then $n_{\text{primer}}$ (blue scale from 0.12 $\mu$m )), and for (c) the penetration depth is calculated with $n_{\text{primer}}$.

beginning of the fine filling part can be detected up to a depth of approximately 0.36 mm, where the noise floor intersects with the A-scan averaging.

The average A-scans depicted in Fig. 9(b) and (c) present the limitation of averaging. As shown in Fig. 9(c), the fluctuations caused by speckles are significant enough to affect the curves fitting if too few A-scans are use in the averaging. For Fig. 9(d), the averaging was performed inside the intentional defect part. In this region we expect a certain amount of topcoat was removed after the intentional defect was made with the needle. In an ideal world, all the topcoat should be removed in this region, but the fitting slop does not correspond to the primer fit in Fig. 9(b). It is possible that there was a non-uniform topcoat residue

above the primer on the 300 A-scans. As we expect to have only primer in an ideal case, the scale for penetration depth was calculated only with $n_{\text{primer}}$.

The purpose of Fig. 10 and Fig. 11 is to demonstrate how the attenuation coefficient can be used to track the topcoat and primer thickness along a B-scan and thus across a WTB. The explanation of the algorithm is given in section 2.3. Although

estimating topcoat and primer thickness in the absence of defects is a straightforward task, the process becomes more challenging in the presence of defects. Figure 10(a) shows the visual estimation of the bottom part of topcoat and primer outside the intentional defect, and then in Fig 10(b), the result of our automatic interface program 20 A-scans averaged for (i) and 100 A-scans averaged for (ii).




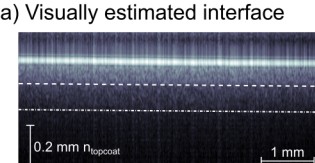
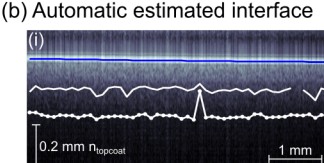
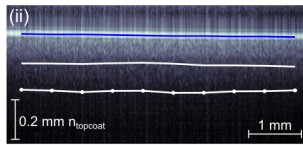

**Figure 10.** (a) Present the estimated thickness of the topcoat and primer with visual inspection. Both lines (from top to bottom) indicate the lower boundary of the topcoat and primer. (b i) and (b ii) present the results of our algorithm for X = 20 and X = 100. The blue line represents the automatic detection of the surface, from top to bottom, while the white lines indicate the lower boundary of the topcoat and primer

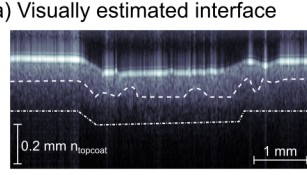
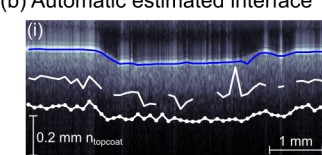
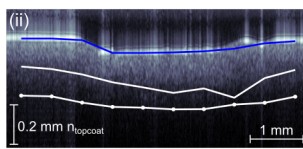

**Figure 11.** (a) Present the estimated thickness of the topcoat and primer with visual inspection. Both lines (from top to bottom) indicate the lower boundary of the topcoat and primer. (b i) and (b ii) present the results of our algorithm for X = 20 and X = 100. The blue line represents the automatic detection of the surface, from top to bottom, while the white lines indicate the lower boundary of the topcoat and primer

Then we applied the program in an intentional defect region. Figure 11(a) shows the visual estimation of the bottom part of the topcoat and primer. As seen in Fig. 9(d), the part where the defect is can be quite challenging to estimate correctly visually. Figures 11(b), the result of our automatic interface program 20 A-scans averaged for (i) and 100 A-scans averaged for (ii). Although the algorithm performs well at the edges of the B-scan, it encounters difficulties in distinguishing each interface in defect regions. In these areas, the slopes of the topcoat and primer layers are sometimes very similar. This issue highlights the potential benefits of integrating algorithms based on deep learning, which could significantly improve the tracking of these two layers' thicknesses.

### 3.2.3 Sub-surface defect

During the manufacturing process of a WTB, some defects can occur as a result of fluctuations in the production processes. These defects may affect the general properties of the coating protecting the blade. From OCT image observations, we categorise defects as: **A Inclusion** local point of high signal (cf. Fig. 12(a,c)); **B Crack** elongated local region of low signal (cf. Fig. 12(b,c)); **C Hole** local point of low signal (cf. Fig. 12(a,b,c)). As the OCT is sensitive to modification of refractive index, the technique is excellent for detecting this category of defects.

Figure 12, presents three typical defects that were detected in sample nº2. In Fig. 12(a), there is the presence of bright particles inside the dashed circle (defect A) and the presence of air bubbles trapped inside the coating (defect C - see white arrows).



Figure 12(b) shows the presence of an air bubble (defect C - left arrow), a crack (defect B - right arrow) just under the area where the intentional defect was made marked by the white arrows, and a hole (defect C - dashed circle). Due to the viscosity of the paint, the bottom part of the layer of the primer or the topcoat may have been moved. Fig. 12(c) shows the presence of
inclusion (defect A - dashed circle), air bubbles trapped inside the coating (defect C - left and middle white arrows), and the suspicion of a crack (defect B - right white arrow).

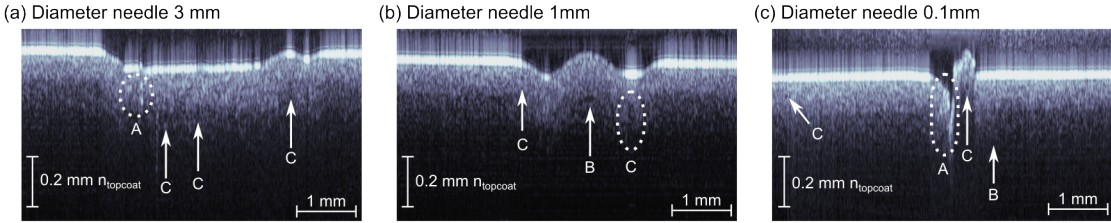

**Figure 12.** Summary of additional defects detected. To enhance visibility, (a-c) represent an average of 5 B-scans. Defect categories found are for (a): A and C, (b): B, and C and (c): A, B and C

## 4 Discussion and conclusion

This research highlights the capability of a MIR OCT scanner to detect internal and surface faults in WTB coatings with
a penetration depth of $\sim 360$ $\mu$m. This depth includes topcoat, primer and beginning of the fine filling layer. Two types of samples were characterised in this study. The first sample was designed to benchmark the penetration depth of the MIR OCT system. An aluminium foil was introduced between each layer during manufacturing. Since metal easily reflects light and gives a signal above the noise floor, this was an effective method to assess whether the MIR OCT system could efficiently penetrate the coating layers, including the topcoat and primer. The second sample was used to test the capability of MIR OCT to detect
manufacturing defects of different categories, from $\mu$m scale to mm scale. An additional challenge with this sample involved tracking the coating layer thickness in B-scans containing fluctuations by analysing the attenuation coefficient. This opens the idea for future research of using AI-based learning techniques to support the characterisation of B-scans, especially to automatically measure the thickness of coating layers.

MIR OCT images were compared to both conventional NIR OCT images and X-ray images. While MIR OCT has improved penetration performance compared to NIR OCT, it is still quite limited when compared to X-ray CT. In comparison, X-ray CT could penetrate the entire coating and GFRP of a blade, in contrary to MIR OCT where the maximum penetration depth is around 360 $\mu$m, which correspond for our case the topcoat and primer layer of the coating. However, MIR OCT outperforms X-ray imaging in a production setting for characterising the WTB coating quality. In using an X-ray CT scanner, it is not
possible to obtain a 3D microscopic map of a WTB region of interest without cutting out a piece of it. This is possible with



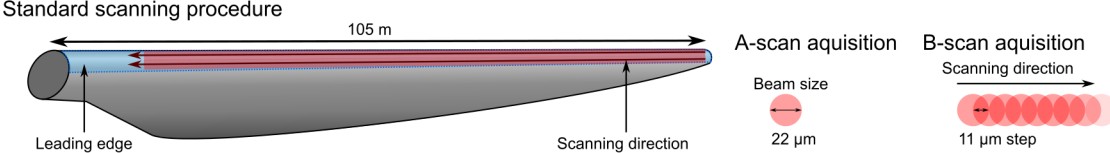

**Figure 13.** Schematic of the proposed standard procedure for blade inspection.

MIR OCT, as the sample characterised does not need to be rotate to extract the data of the volume. Additionally, the resolution provided by the OCT measurement is better than the one provides by X-ray CT.

MIR OCT scanner has the potential of replacing the local and destructive dolly test in the production and complementing
existing methods by adding structural information on the microscopic level, which is not possible with other techniques. In addition to side-view information, it can generate a full topography map, topcoat and primer thickness maps, bulk uniformity of the coating, and, importantly, a count of the number and types of defects. It is important to minimise the number of defects inside the WTB coating. Indeed, initial defects play a role in the erosion and speed up the fatigue. For example, air bubbles trapped inside the coating create stress concentration and will intensify the damage area from water droplets (Fæster et al.
(2021); Mishnaevsky et al. (2021)).

An extraordinary insight would be obtaining the above-mentioned quality measures for a full-length WTB scan, such a scan is illustrated in Fig. 13. Currently the scanning time is approximately 40 min for a $5.73 \times 5.73$ mm$^2$ area. It is interesting to consider potential impact of a custom scanner developed WTB screening. If a production-tailored MIR OCT scanner can
achieve a ten-fold improvement in speed, a scan of the full blade length of 105 m can be done within 40 minutes.
In conclusion, this study demonstrated that the MIR OCT system can track the impact of intentional defects introduced during manufacturing. The results are encouraging for using MIR OCT as a complementary tool to other NDI techniques in quality testing the full WTB volume.

*Data availability.* The data presented in this study are available on request from the corresponding author. The data are not publicly available
due to significant storage requirements.

*Author contributions.* Conceptualisation: C.L., C.P., O.B., N.I. Data curation: C.L., C.P., J.B, S.F, O.B., N.I. Formal analysis: C.L., C.P., N.I. Funding acquisition: O.B., N.I. Investigation: C.L., S.F. Methodology: C.L., C.P., S.F., N.I. Project administration: O.B., N.I. Resources: P.N., T.W. Software: C.L., C.P., N.I. Supervision: O.B., N.I. Validation: C.L., O.B., N.I. Visualisation: C.L. Writing (original draft preparation): C.L. Writing (review and editing): C.L., C.P., P.N., T.W., J.B., S.F., O.B., N.I. All authors have read and agreed to the published version of
the manuscript.



*Competing interests.* The authors declare that they have no known competing financial interests or personal relationships that could have appeared to influence the work reported in this paper.

*Acknowledgements.* We would like to acknowledge María Rocío Del Amor, Fernando García Torres, Natalia Lourdes Perez Garcia de la Puent, Adrian Colomer Granero and Prof. Valery Naranjo of the CVB lab, la Universitat Politècnica de València for rewarding discussions on localization and annotation of material defects observed in OCT images. We also thank the TURBO consortium for useful discussions on scope and applications of MIR OCT as an NDI technology.





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
