# Peer review of "Non-destructive sub-surface inspection of multi-layer wind turbine blade coatings by mid-infrared Optical Coherence Tomography"

_Wind Energy Science, 2025_

## Referee Comment (RC2)

**Confidential Comments to the Editor:**

Thank you for the opportunity to review the Manuscript wes-2025-237 entitled " **Non-destructive sub-surface inspection of multi-layer wind turbine blade coatings by mid-infrared Optical Coherence Tomography**" for the Journal of **Wind Energy Science**.

To the best of my knowledge, I have thoroughly reviewed the manuscript, and I would recommend in **revise the manuscript with major comments** based on my review.

Thanks again for providing the opportunity to contribute the review support to maintain the quality standard of **Wind Energy Science** Journal.

I look forward to contributing more review support, so please share any additional manuscripts within my area of expertise. Thank you!

**Comments to the Author:**

As reviewer of the Manuscript wes-2025-237 entitled " **Non-destructive sub-surface inspection of multi-layer wind turbine blade coatings by mid-infrared Optical Coherence Tomography**", I have thoroughly reviewed the manuscript, and I would recommend addressing the below comments to make the study more wholistic nature to appreciate the NDI of multi-layer wind turbine blade coatings by MIR OCT Technique:

1. Under Abstract section, there should be executive summary of the quantitative conclusions of the proposed research. The abstract should be rephrased to make it more wholistic of the complete study on the manuscript.
2. Considering the inspection depth upto two layers, what structural advantages can be accomplished as initially is quoting that the reducing the quantity of waste and thereby cost of production. Please add justification for the proposed study?
3. What is the capability of the described NDI techniques application on field blades compared to conventional inspection techniques.
4. What is the threshold capability of inspection depth for the mentioned MIR OCT technique.
5. Apart from the introducing the application of MIR OCT technique to wind blades from other applications, what are the extended research capabilities addressed in the proposed manuscript?

I hope my critique helps the authors to improve their work and find useful in this review. Thank you!

---

## Author Comment (AC1)

Dear reviewer,

Thank you for your time and effort spent into the review of our manuscript. We hope this reply sufficiently answers your questions.

1. Penetration depth, defect detectability, and layer-thickness extraction are treated mainly qualitatively. Quantitative rigor is insufficient in several key analyses. The study would benefit from statistical summaries over multiple locations and more formal error metrics.

Thanks for your comment. For the maximum penetration depth, this was calculated through the comparison of the noise floor of our OCT system and the measurement of the sample outside any intentional defect. The noise floor was measured when there is no sample under the head scan, as presented in figures 8 and 9. Penetration depth is intrinsically dependent on the system combined with the chosen sample, representation one layer composition example applied within the wind turbine blade industry.

In addition, as all OCT B-scans contain speckle noise, it was necessary to compare our measurements with the noise floor of our system to ensure rigorous validation to distinguish between scattering from the material and sample enabled speckle patterns.

2. The attenuation-based interface-detection algorithm seems underdeveloped. The conceptual description (Fig. 5) is helpful, but the lack of methodological detail, no validation against ground truth, and poor performance in defected areas weaken the Conclusions. Suggested claims about future integration of deep learning are reasonable, but they should not compensate for the current missing quantitative evaluation.

About the algorithm, we agree; this algorithm is very basic and follows the attenuation slopes of the different layers.

The validation slope parameters (*objective fits*) have been measured outside the intentional defect area with a large quantity of data averaged to ensure the slope value extracted was not influenced by any noise or manufacturing defect. The data size represents 100 B-scans composed of 1000 A-scans each, so the slope calculation for the objective fits is based on $10^5$ mean A-scans.

Then these *objective fits* have been used in the algorithm to find the layer over the B-scan studied.

This algorithm has been made to help us to isolate the different layers more easily, but as it is a primitive version, we still see room for improvement, e.g. by increasing the dataset size. About the deep learning integration, this one is already in progress with our project partner. We would like to implement deep learning solutions to automatise the layer thickness detection process during manufacturing. Combining larger datasets and knowledge transfer, we are working on improving the precision of coating layer interface delineation.

3. Industrial scalability claims seem overstated. The speculative calculation about faster scanning times lacks a realistic time-budget analysis and seems optimistic. These observations should be presented more cautiously, experimentally backed up, or omitted.

Thanks to the reviewer for this comment. Yes, indeed, the scalability seems to be not feasible with the speed value given from the experimental one. We dare extrapolate on our results as we are collaborating with NORBLIS (https://norblis.com/)developing industrial MIR OCT scanners suited for industrial demands. A LinkedIn post has been published for a milestone on this development (https://shorturl.at/VQJOw).

4. Key procedures, including refractive-index derivation, OCT post-processing, and sample preparation, need clearer and more rigorous descriptions to allow replicability.

The refractive indexes have been calculated through the conventional method as explained on page 8. Using the sample with aluminium foil inside. Optical path delay (OPD) in the air represents the physical distance, as the refractive index of the air is approximately equal to 1 for a temperature of 21 degrees.

We extracted the physical distance $L_{topcoat}$ and $L_{topcoat+primer}$ by removing the layer above the corresponding aluminium foil and measured it through our NIR OCT B-scans. Then, as it was possible to identify these two aluminium foils with the MIR OCT system, we calculated the corresponding refractive index.

The OCT post-processing is explained in this article https://doi.org/10.1038/s41377-019-0122-5 as presented inside the article in material and method.

About sample preparation, they were made by Siemens Gamesa, and we are not allowed to share the detailed procedure of it.

5. Related to the above, the description of the sample preparation requires greater detail.

This is a good suggestion, but unfortunately, as these samples were made by Siemens Gamesa we cannot provide many more details.

6. The state-of-the-art for damage assessment in GFRP is relatively limited. Many recent contributions can be found for this specific application, for example, in the field of vibration-based monitoring, https://doi.org/10.1016/j.compstruct.2020.112882 and https://doi.org/10.1002/stc.2805. These could be added to the discussion in the Introduction.

Thanks for your notice, as the paper focuses on the characterisation of the coating above the GRFP of the wind turbine blade (WTB) because the state-of-the-art for damage assessment in GRFP is poor. The idea here was mainly to provide a quick representation of how the different non-destructive techniques are used in WTB. But we will cite and include the most recent article of the two proposed.

7. The text repeatedly highlights penetration of "~360 μm". However, it seems that this value corresponds to a single dataset (sample no. 1) and thus is not a generally valid statement. It is one value of one measurement. The authors should provide statistics across several scans.

For the maximum penetration depth, this one was calculated through the comparison of the noise floor of our OCT system and the measurement of the sample outside any intentional defect. This value is the maximum that we could obtain. For other similar samples we screened, the penetration with slightly lower but beyond 300 μm .

8. The figures could benefit from higher contrast and more explicit annotation

Thanks for the suggestion, but the figures are already very saturated by the annotation. By adding more, we are afraid that we will compromise the reading and the comprehension of the manuscript.

9. The English is overall good, but there are occasional grammatical errors, such as missing articles. A thorough review could be useful.

Thanks for noticing this; the latest typos have been corrected.

---

## Author Comment (AC2)

Dear reviewer,

Thank you for your time and effort spent into the review of our manuscript. We hope this reply sufficiently answers your questions, we also updated our manuscript where it was necessary.

1. Under Abstract section, there should be executive summary of the quantitative conclusions of the proposed research. The abstract should be rephrased to make it more wholistic of the complete study on the manuscript.

Thanks for the comment. The abstract updates :

Non-destructive inspection (NDI) is useful in the industrial sector to ensure that manufacturing follows defined specifications, reducing the quantity of waste and thereby the cost of production. Optical Coherence Tomography (OCT), a well-known diagnostic technique in medical and biological research, is increasingly being used for industrial NDI. In the mid-infrared (MIR) wavelength range, OCT can be used to characterise parts and defects not possible by other industry-ready scanners, and enables better penetration than conventional near-infrared OCT.

In this article, we demonstrate NDI of wind turbine blade (WTB) coatings using an MIR OCT scanner employing light around 4 μm from a supercontinuum laser source. We inspected the top two layers of the coating (topcoat and primer) in two different samples. The first is to determine the maximum penetration depth, and the second one is to imitate defect identification. We also developed a basic algorithm to extract the thickness layer of the topcoat and primer. The results of our study confirm that MIR OCT scanners are promising for coating inspection and quality control in the production of WTBs, with performance parameters not achievable by other technologies.

2. Considering the inspection depth upto two layers, what structural advantages can be accomplished as initially is quoting that the reducing the quantity of waste and thereby cost of production. Please add justification for the proposed study?

The focus of this study was limited to the development of the coating layer inspection performance and potential. However relevant, investigations of the structural is out of the scope of this work.

3. What is the capability of the described NDI techniques application on field blades compared to conventional inspection techniques.

This NDI technique has the capacity to do subsurface characterisation and 3D visualisation without damaging the material used inside the coating, as there is no radiation emitted by the source like X-rays or UV. This also provides a safe working environment for the user.

The sample is characterised as normal at the surface of it and does not require any rotation of the sample to have access to the 3D volume information. For a large system like a blade, this reduces the complexity of the characterisation, and in principle can be expended to the field with proper amount of engineering. We are collaborating with the company NORBLIS (Norblis.com) which is looking into the engineering challenges of MIR OCT scanner usability and deployment.

The resolution of the OCT scanner is intrinsically linked to the OCT system (the light source used and the optics inside the head scan) and not to the sample size. In summary, the MIR OCT system will be able to characterise with the same finesse an electronic chip and a wind turbine blade.

4. What is the threshold capability of inspection depth for the mentioned MIR OCT technique.

The capability of the system is the one presented ~360 μm of penetration depth, but we noticed several months after having done the experiment that our upconversion system had a problem. The conversion

efficiency dropped by a factor of 2 after seven years. The module is currently in reparation. With this the penetration depth might be even better than the reported.

5. Apart from the introducing the application of MIR OCT technique to wind blades from other applications, what are the extended research capabilities addressed in the proposed manuscript?

The MIR OCT scanner has the potential to characterise various nonorganic materials, such as ceramics, paper and electronics components. The wavelength of the source used allows a better penetration than conventional OCT (with NIR wavelength) due to a reduction of scattering and so of the general losses.